# The Trade-Off between the Controller Effort and Control Quality on Example of an Electro-Pneumatic Final Control Element

**Michał Bartyś** *[ID] and **Bartłomiej Hryniewicki**

Institute of Automatic Control and Robotics, Warsaw University of Technology, św. A. Boboli 8,
PL 02-525 Warsaw, Poland; 258971@pw.edu.pl
* Correspondence: bartys@mchtr.pw.edu.pl; Tel.: +48-501-697-545

**Abstract:** For many years, the programmable positioners have been widely applied in structures of modern electro-pneumatic final control elements. The positioner consists of an electro-pneumatic transducer, embedded controller, and measuring instrumentation. Electro-pneumatic transducers that are used in positioners are characterized by a relatively short mean time-to-failure. The practical and economical method of a reasonable prolongation of this time is proposed in this paper. It is principally based on assessment and minimizing the effort of the embedded controller. For this purpose, some measures were introduced: The control value variability, mean-time, and the cumulative controller's effort. The diminishing of controller effort has significant practical repercussions because it reduces the intensity of mechanical wear of the final control element components. On the other hand, the reduction of the cumulative effort is important in the context of process economy due to limitation of the consumption of energy of compressed air supplying the final control element. Therefore, the minimization of control effort indicators has an impact on the increase of the functional safety and economics of the controlled process. The simulations were performed in the Matlab-Simulink environment with the use of the liquid level control system in which a phenomenological model of a final control element was deployed. As a result of the performed simulations, the recommendations regarding the selection of the structure of positioner controller were elaborated.

**Keywords:** final control element; electro-pneumatic transducer; controller effort; control quality factors; wear; mean-time-between-failures

---

## 1. Introduction

In the structures of the closed loop industrial automation systems, we can generally distinguish: Controlled systems, controllers, measuring instrumentation, and final control elements [1–3]. The general structure of a single-loop control system is depicted in Figure 1. The final control elements are physical units directly affecting the streams of energy and materials. In the structure of a control system, they simply play the role of adapters between controllers and controlled systems.

In fact, the final control elements are acting as energy or power transformers that convert the low-energy or informative control value (CV) into a high-energy driving output. Clearly, due to the principle of energy conservation, the final control elements require an additional auxiliary power supply source.

The main interest of this paper is focused on the certain class of final control elements in which the compressed air is used as an auxiliary energy supply source. These elements are commonly used in automatic control systems in the following industries: Power, chemical, petrochemical, pharmaceutical, and food.

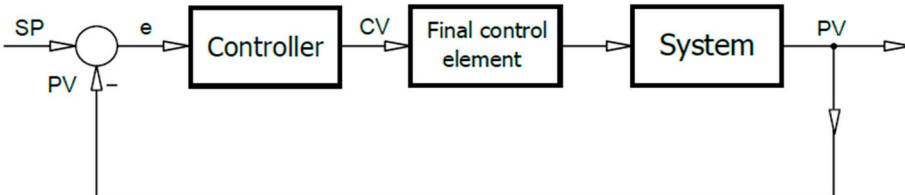

**Figure 1.** Simplified block diagram of the structure of the closed loop automatic control system. Notion: SP—setpoint; PV—process variable; e—control error; and CV—control value.

A typical electro-pneumatic final control element consists of a pneumatic actuator with linear or rotational movement of its stem, a positioner, and a control valve (Figure 2). By adjusting the position of the valve plug attached to the end of the actuator's stem towards the valve seat, it is possible to control the flow rate of a medium passing through the valve.

The primary goal of the positioner is to follow-up the CV. In order to do it, the positioner has to control the pressure of the pneumatic actuator in such a way that the position of the valve plug will depend exclusively on CV and will suppress the influence of the disturbances such as: Changes of operating temperature, supply pressure fluctuations, changes in the static and dynamic load of the actuator's stem, friction, and the evolution of friction forces.

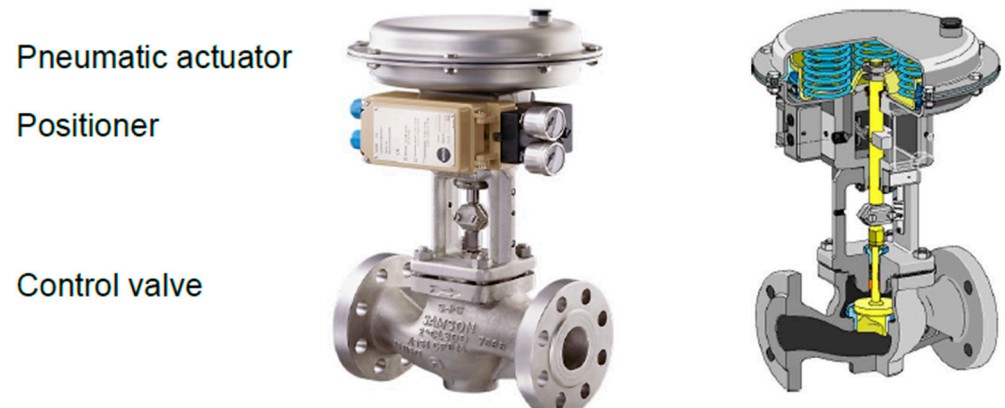

**Figure 2.** An example of an electro-pneumatic final control element and cross-section of its mechanical construction [4,5].

The final static and dynamic properties of the actuator are shaped by the positioner. In fact, the positioner is a specialized, autonomous control system of the stem position. The simplified block diagram of the final control element is shown in Figure 3.

When applied, one requires, from the final control element, a repetitive non-hysteresis static characteristics, aperiodic step response, and minimal control time. In principle, realization of these requirements becomes realistic only due to the use of a positioner.

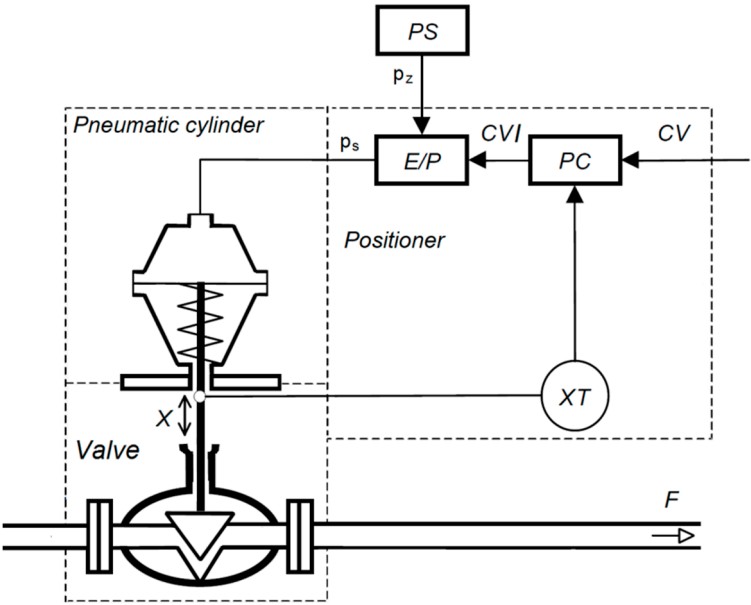

**Figure 3.** Block diagram of an electro-pneumatic final control element. Notions: CV—output of external controller; CVI—output of internal controller; PC—internal controller; E/P—electro-pneumatic transducer; PS—compressed air supply; $p_z$—supply air pressure; $p_s$—the output pressure of the E/P transducer; X—displacement of the valve plug; and XT—displacement transducer of the actuator's stem [6].

It should be noted that the final control elements are exposed to adverse environmental conditions such as thermal shocks, a wide range of operating temperatures, vibrations, humidity, dusty pollutants, corrosive working environment, and electromagnetic interference. Therefore, the final control elements are classified as belonging to the group of elements of automation systems subjected to the most frequent failures. It is estimated [7] that the probability of the failure of actuator may be even at least one order higher than for sensors.

The faulty final control element can lead to a deterioration of the quality of the final product, and can even lead to the process shutdown. All these factors influence the safety of the process and humans involved, as well as worsen the economic indices of the process. For this reason, the assurance of the most trouble-free operation of such devices becomes more important. In this context, the issues of fault prediction and diagnostics, as well as ways of prevention actions and fault tolerant control become particularly pertinent [8]. Most frequently, the electro-pneumatic transducer is subject to fail in the case of electro-pneumatic final control elements. This component fails due to poor quality of supply air, friction wear of its mechanical components, and material fatigue of its moving parts.

In this paper, a practical and economic approach to prolongation of the final control element lifetime is proposed. It is based on the appropriate shaping and tuning of the positioner controller in such a manner that it results in the limitation of the amplitude and of the number of cycles on its output. This directly influences the wear and fatigue of moving elements of the electro-pneumatic transducer. This type of approach, due to the conclusions drawn from the Wöhler's fatigue curve [9], has significant impact on mean-time-between-failures (MTBF), and therefore, reduces wear and increases functional safety of the controlled process.

The essence of the proposed approach is to change the structure and parameters of the positioner controller (PC) in order to minimize its effort, but not at the expense of worsening the control quality factors of the controlled system in which such an element is used.

The structures of positioner controllers of temporary final control elements are programmable as a rule. In order to achieve sufficiently short sampling time, the implementation of the controller algorithm should be time effective. This is not a trivial task when taking into account strong limitations

of truly disposed computational power. Therefore, the structure of the controller should not be too complex. This paper tries to answer how to choose the controller's structure from the limited set of arbitrary chosen low complexity potential positioner's controllers in order to keep the desired control quality factors by minimal effort of the controller.

The results of research of the proposed approach were obtained in the Matlab-Simulink simulation environment with the use of the final control element model [10], which is recognized in the process fault detection and isolation (FDI) society. This is, to some extent, enhances trustworthiness of presented results.

## 2. The Controller Effort

The term controller effort is not novel, however, may be, undeserved discussed relatively very rare. Moreover, the term controller effort does not have any unambiguously established definition. Some papers refer to this term, however, leaves it undefined [11,12]. In these papers, controller effort is understood as controller output. In turn, in Reference [13], the control effort is defined as the instantaneous value of the controller output. In Reference [14], the term actuator effort is defined quite differently, as connected with a physical value of a force. This rather complies with the generalized term of effort known from the bond graph theory. In the papers from References [15,16], the control effort is defined as an integral part of the power of a control signal. Therefore, this definition can alternatively be interpreted in terms of energy of the control signal. Alternately, Beshi et al. in [17] verbally defined the control effort as root-mean-square value of the control signal increments. This definition may be interpreted as penalizing averaged changes of the energy of the control signal velocity when applied to the optimization problem. The control effort index defined in Reference [18] reflects the control energy usage normalized by the energy of the raw reference signal. In fact, the introduced index characterizes the output-input signal energy effectiveness of the controller. However, from the practical point of view, this definition is less important, because the instantaneous consumption of energy in real systems is rather a function of control signal.

For the sake of this paper we introduce a slightly different definition of the controller effort together with some associated terms referring to the variability of the control signal. These outcomes from a heuristic rule connect the lifetime of the mechanical component with the number of working cycles. Further, the definitions will be given for the continuous and discrete time domain systems. Definitions in a discrete time domain are useful for real implementations in a digital environment.

**Definition 1.** *Define the normalized variability of the controller output* $V(t)$ *as:*

$$V(t) = \frac{1}{\Delta v}\left|\frac{dv(t)}{dt}\right|, \tag{1}$$

*where:* $\Delta v = |v_{max} - v_{min}|$—*nominal range of controller output.*

**Definition 2.** *Define the controller's mean time effort over time interval* $\Delta t$ *as an average normalized variability of the controller output.*

$$Q(\Delta t) = \frac{1}{\Delta t}\int_{t_0}^{t_1} V(t)dt = \frac{1}{\Delta t}\frac{1}{\Delta v}\int_0^t\left|\frac{dv(t)}{dt}\right|dt, \tag{2}$$

*where:* $\Delta t = (t_1 - t_0)$. *By changing integration limits in Equation (2), we can define cumulative time effort:*

**Definition 3.** *Define the controller's cumulative effort as an averaged over time normalized variability of the controller output.*

$$Q(t) = \frac{1}{t}\int_0^t V(t)dt = \frac{1}{t}\frac{1}{\Delta v}\int_0^t\left|\frac{dv(t)}{dt}\right|dt. \tag{3}$$

In the discrete time domain, the normalized variability can be expressed as:

$$V(k) = \frac{1}{\Delta v} \left| v_k - v_{(k-1)} \right|,$$ (4)

where: $v_k$—$k$-th sample of the controller output $v(t)$. By analogy to Equation (2), the mean time effort in the discrete time domain can be expressed as the averaged sum of the variability of control signal over the number of $\Delta k$ samples:

$$Q_{\Delta k} = \frac{f_s}{(\Delta k - 1)} \sum_{k=k_o+1}^{k_1} V(k) = \frac{1}{\Delta T} \frac{1}{\Delta v} \sum_{k=k_o+1}^{k_1} \left| v_k - v_{(k-1)} \right|.$$ (5)

where: $\Delta k = (k_1 - k_0)$; $f_s$—sampling frequency of controller output $v(t)$; $\Delta T = \frac{f_s}{(\Delta k - 1)}$—time interval.

**Remark.** *The introduction of sampling frequency $f_s$ in Equation (5) avoids deflating/inflating effects of the mean time effort for different sampling frequencies.*

Finally, by analogy to Equation (3), the cumulative effort in the discrete time domain can be expressed as the averaged sum of the variability of control signal over the *K* samples:

$$Q_K = \frac{f_s}{(K-1)} \sum_{k=1}^{K-1} V(k) = \frac{f_s}{(K-1)} \frac{1}{\Delta v} \sum_{k=1}^{K-1} \left| v_k - v_{(k-1)} \right| = \frac{1}{T} \frac{1}{\Delta v} \sum_{k=1}^{K-1} \left| v_k - v_{(k-1)} \right|,$$ (6)

where: $T$—time horizon.

In practice, in automation systems, the control value is usually normalized. In this case: $\Delta v = 1$, and therefore respective Formulas (1)–(6) appropriately simplify it.

If we further assume that the movable elements of the electro-pneumatic transducer of positioner approximately reproduce the trajectory of the control signal, then the reduction of the controller's effort leads to a diminishing of the average amplitude and the totalized travel of its moving elements. As a result, it is expected to reduce the intensity of the wear of its mechanical components and thus extends MTBF. Figure 4 presents an example of two control strategies for the electro-pneumatic transducer: An aggressive—marked with a red line and a conservative marked by the dark blue line. The appropriate cumulative effort values for both strategies in the time horizon of $\Delta t = 10$ s differ significantly, and respectively, are equal to: 0.32 and 0.029. In this case, the eleven-fold reduction in the controller's effort allows the increase of the permissible number of work cycles due to the significant decrease of their amplitude.

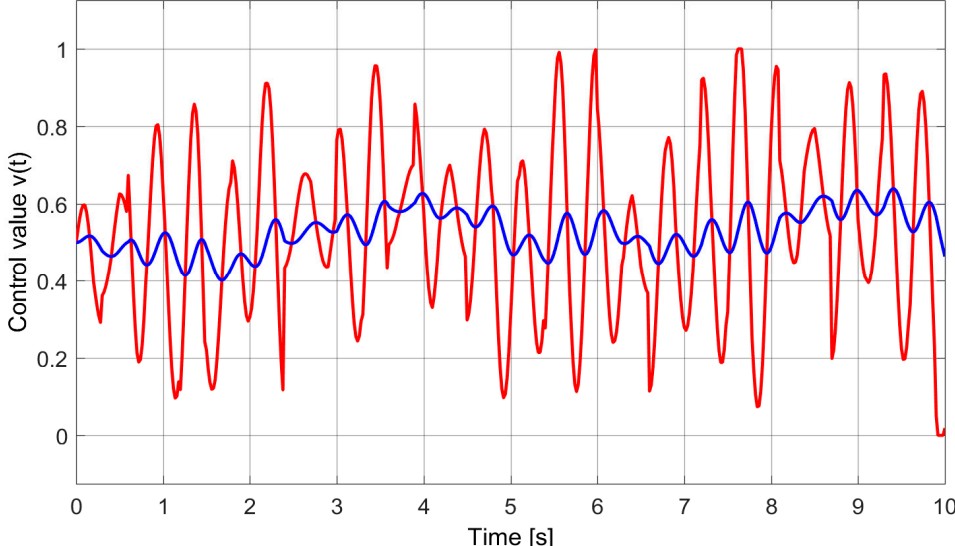

**Figure 4.** An example of two control strategies with a strongly differentiated effort. Value of effort for aggressive strategy $Q_K = 0.32$ and for a conservative strategy $Q_K = 0.029$ [19].

## 3. Research Environment

The implementation of experimental investigations in a real life scenario is usually costly and time consuming. For this reason, the choice of the structure, parameters, and tests of the proposed control strategy was carried out in a simulation. For this purpose, a complex, phenomenological model of an electro-pneumatic final control element was used. This model was prepared and validated especially for the assessment model based fault detection and isolation approaches [20].

The choice of this model was motivated by its availability [10] and the recognition in the international community of process safety diagnostics. Simulation tests were performed by means of the liquid level control system in which the final control element from Reference [10] was applied. A simplified block diagram of the simulated control system is shown in Figure 5.

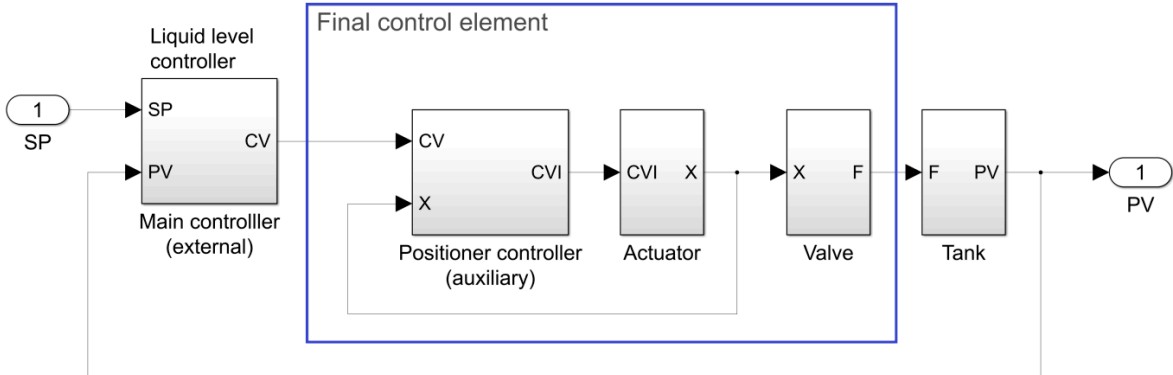

**Figure 5.** Simplified simulation model of the liquid level control system. Notions: SP—liquid level setpoint; PV—measured liquid level; CV—output of the main process controller; CVI—output of the internal positioner controller; X—position of the valve stem; F—liquid flow rate; and PV—liquid level in the tank.

The constant set-point liquid level control system was taken from the DABlib library [6] in order to provide the near realistic and traceable conditions. Some comments regarding the adopted model referred to in Reference [6], as a Simple Process Example IV, are given below.

The liquid is stored in a cylindrically shaped tank having constant cross-sectional area $A = 20$ m$^2$. The liquid level is measured by means of level sensor. Level signal is fed back to the main proportional-integral controller ($k_p = 10$, $T_i = 50$ s) which governs the final control element. The liquid is free discharged from the tank. The liquid outflow is modeled by the equation:

$$F_{out} = \mu \cdot \sqrt{2 \cdot \rho \cdot g \cdot h}, \tag{7}$$

where $\mu = 0,13$—flow contraction coefficient; $\rho = 1$kg/dm$^3$—specific gravity of the liquid; $g = 9.81$ m/s$^2$—gravitational constant; and $h$—liquid level in [m].

The liquid level is described by the following equation:

$$h = \frac{1}{A} \int (F_{in} - F_{out}) dt, \tag{8}$$

where $F_{in}$—liquid inflow into tank.

The inflow liquid is supplied from a positive displacement pump. The static pressure generated by the pump forcing liquid is equal to 3.5 MPa. The pressure drop in the supply pipeline is simulated by a constant hydraulic resistance equal to 1 kPa/m$^3$/s. The final control element is intended for controlling liquid inflow to the tank.

A simplified block diagram of the simulated final control element is depicted in the Figure 6. The detailed model of the final control element is available if we look under the mask of the block ACT from the DABlib Simulink library [10]. The usage of the model is described in the document Using Damadics Actuator Benchmark Library (DABLib) available from the same site.

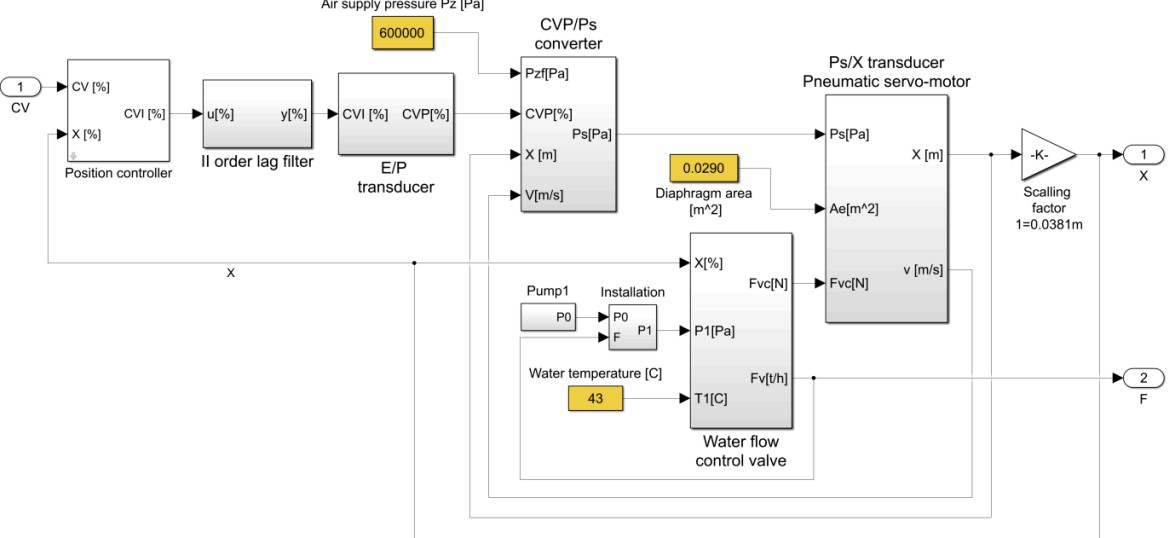

**Figure 6.** Simplified block schematics of the Simulink model of the final control element.

The second order lag filter visible in Figure 6, and following the positioner controller, represents dynamics of the electro-pneumatic transducer E/P. This filter is not a part of the positioner controller.

Applied final control element exhibits non-linearity, ambiguity, and asymmetry of the static and dynamic characteristics (Figures 7 and 8).

It should be noted that the directionality of dynamic characteristics is a major challenge for the internal positioner's controller and has a significant impact on the variability of the controller's effort.

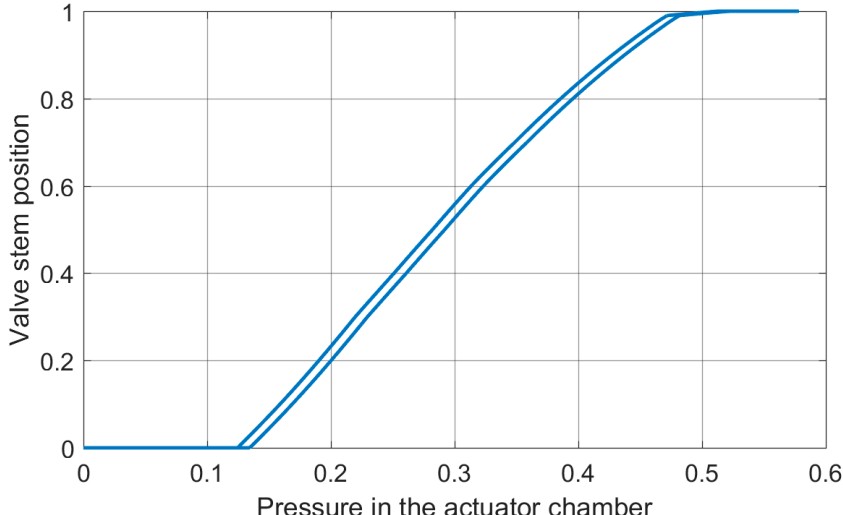

**Figure 7.** Normalized static characteristics of pneumatic actuator of the electro-pneumatic final control element.

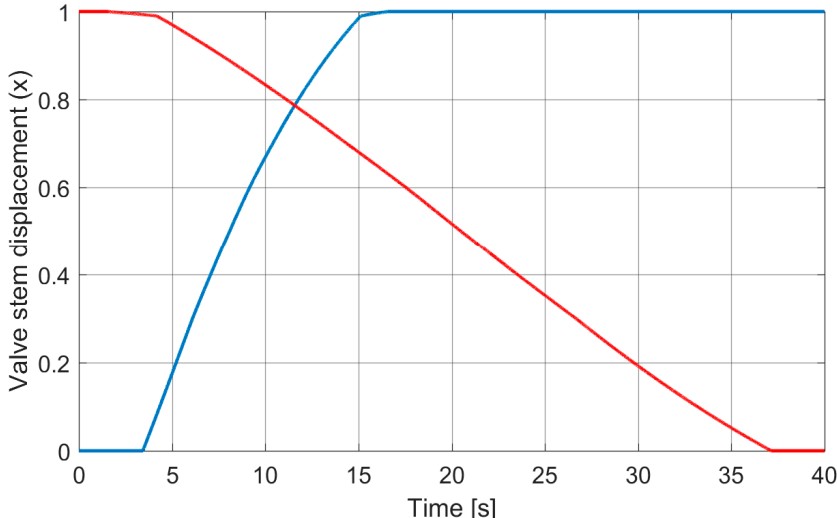

**Figure 8.** Positive (blue line) and negative (red line) step response of the valve stem displacement of pneumatic actuator.

## 4. Control Quality Factors

The controller effort is just one of the many control quality indices used in order to evaluate the features of automation systems [1–3]. Therefore, the minimizing effort of the controller cannot be separated from the analysis how this influences other control quality factors. There will be seven control quality indicators analyzed in order to obtain a more complete view of the effects of minimizing the effort of the internal controller of the positioner:

- cumulative effort of the controller $Q_K$ according to formula (6) for $T = 100$ s;
- normalized, average absolute tracking error $e_{Kr}$ according to the formula (9). The value of this factor is determined in test when applying a standardized trapezoidal setpoint shape with the constant slope equal to 0.025 s$^{-1}$.

$$e_{Kr} = \frac{f_s}{K} \sum_{k=1}^{K} |CV(k) - X(k)|, \tag{9}$$

where $X$—positioner stem displacement;

- normalized, average absolute tracking error $e_{Ks}$, according to Formula (9). The value of this factor is determined in a test in which a rectangular set-point with the 50% amplitude and period equal to 40 s is applied.
- overshoots $\kappa_{rise}$ and $\kappa_{fall}$ obtained respectively for applying positive and negative 60% stepwise set-points. Overshoot is defined as the ratio of the amplitude of the first transitional control error $e_1$ to the setpoint change $e_0$ and is expressed as a percentage.

$$\kappa = \frac{e_1}{e_0} \tag{10}$$

- settling times: $T_{Rrise}$ and $T_{Rfall}$ for the 60% stepwise setpoint changes appropriately in positive and negative direction settling time is defined as the time that elapses from the moment of the set-point change until a positioner's stem position X settles within $\pm 0.05 e_0$ tolerance band around the steady state value.

## 5. Positioner Controller

Without any doubt, the structure and parameters of internal controller of positioner influences control quality factors as well as control effort. This paper presents briefly the results of an experimental selection of the structure and parameters of the internal controller of the positioner applied in the control system shown in Figure 5.

As noted in Reference [21], choosing the controller for a given control problem is not a trivial task. Cit. "This choice is often made heuristically by the design engineer." This problem is exhaustively exemplified in [22]. This paper tries to answer how to choose the structure of positioner controller from a certain limited set in order to maintain desired plant control quality factors and simultaneously minimizing controller effort.

*5.1. Methodology*

The limited set of four arbitrary chosen controller structures were examined: proportional (P), proportional-derivative (PD), fuzzy PD (FPD) and neural network (NN). The set of chosen controllers is not exhaustive but sufficient enough for exemplification of the issue of control effort. Integral action of controllers was not considered here because of the system stability problem and therefore necessity in assuring sufficient gain and phase margins.

Firstly, the necessary and sufficient conditions regarding comparability of achieved results are formulated.

- the all controllers should be applied successively in the same control system. This allows a meeting of the necessary condition of comparability.
- the settings of all investigated controllers should be optimized by means of the same cost function. This satisfies the sufficient condition of comparability.

The chosen ad hoc cost function (11) reflects the weighted linear combination of desirable properties of the positioner, i.e., some trade-off between tracking error and controller effort.

$$J = \frac{f_s}{K} \sum_{k=1}^{K} |CV(k) - X(k)| + \gamma \cdot \frac{1}{T} \frac{1}{\Delta v} \sum_{k=1}^{K-1} \left| v_k - v_{(k-1)} \right| \tag{11}$$

where $\gamma$—is the weight of the controller effort. The weight $\gamma = 30$ was finally chosen for experiments.

Here, the question appears why do not construct the cost function in such a manner that it will allow for minimizing of the positioner controller effort instead of minimizing its tracking error? While this expectedly leads to solution where the optimal effort would be close to zero and therefore does not have practical sense.

Next, for all investigated controller structures there were calculated values of all quality factors presented in Section 4. Obtained results are shown in Table 3.

### 5.2. Proportional and Proportionl-Derivative Controller

The settings of classic linear proportional (P) and proportional-derivative (PD) controllers are shown in Table 1.

**Table 1.** The nominal settings of P and PD controllers.

| Controller | $k_p$ | $T_i$ [s] | $T_d$ [s] |
|:---:|:---:|:---:|:---:|
| P | 100 | $inf$ | 0 |
| PD | 100 | $inf$ | 0.244 |

The values of proportional gain for both controllers were limited to 100.

Despite the high proportional gain, the final control element behaves stable. High proportional gain allows for obtaining low tracking errors and fast responses to disturbances. Thus, the resulting equivalent stiffness of the final control element is beneficially high.

### 5.3. Fuzzy PD Controller

The development of fuzzy nonlinear controllers became a mile-stone for the applied fuzzy logic theory. Early fuzzy controller solutions can be found in References [23,24]. A proportional-derivative fuzzy controller with error and error derivative inputs was presented in [25]. Its simple structure allows for easier optimization and reduction of the required computational power. Therefore, this type of controller is extensively exploited in research reported in this paper. The development of the new optimization approaches for fuzzy controllers [24,26] allows for improvement of the quality of fuzzy controllers. Interesting evidence of the equivalence of the classic PID and Mamdani fuzzy controllers can be found in Reference [27]. Over the years, new and better structures of fuzzy controllers were developed [28–30], but their implementation in a positioner controller could be too burdensome.

The structure of the fuzzy PD controller used in this research is depicted in Figure 9. The fuzzy controller has two inputs (control error and derivative of the control error) and one control value output. The three linguistic values (membership functions) were associated with each input. The shapes of input membership functions are depicted, respectively, in Figure 10a,b. Further, the five singleton membership functions visible in Figure 10c were associated with the controller output. This allows for using fast center of gravity defuzzification algorithm of the aggregated fuzzy conclusion generated by the nine rules shown in Table 2.

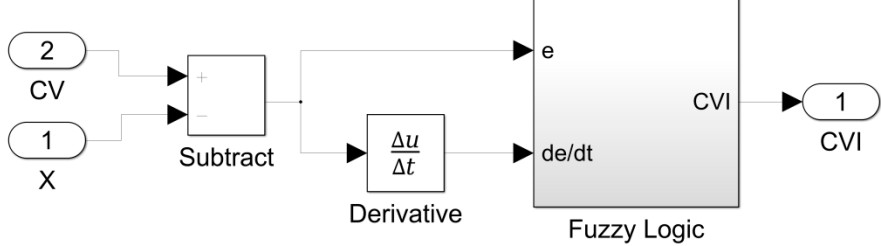

**Figure 9.** The simplified block schematics of the Simulink model of the fuzzy PD controller.

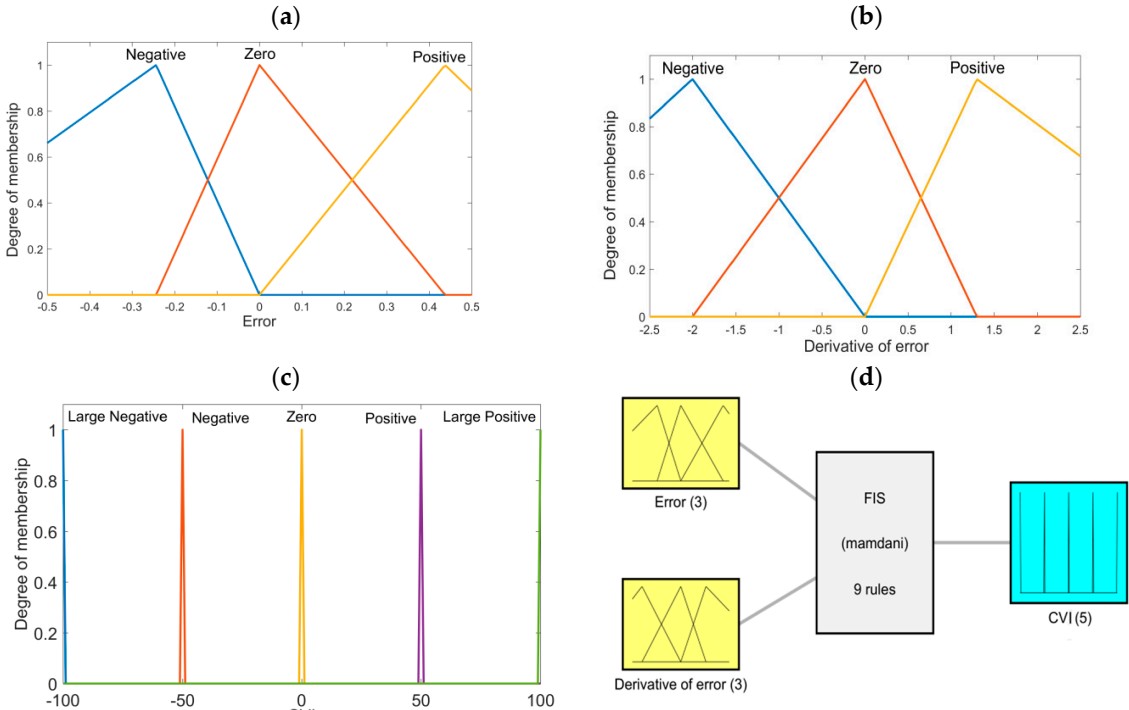

**Figure 10.** (**a**) Membership functions representing fuzzy values of the control error. (**b**) Membership functions representing fuzzy values of the derivative of the control error. (**c**) Membership functions representing the fuzzy control value output. (**d**) Block diagram of the fuzzy reasoning applied for fuzzy PD controller.

The fuzzy reasoning is based on standard skeleton base rule depicted in Table 2. It counts together nine rules. The classic Mamdani's fuzzy inference machine is applied for fuzzy reasoning.

**Table 2.** The base rule of the fuzzy PD controller.

| Control Error | Derivative of Control Error | Output |
|---|---|---|
| Negative | Negative | Large Negative |
| Negative | Zero | Negative |
| Negative | Positive | Zero |
| Zero | Negative | Negative |
| Zero | Zero | Zero |
| Zero | Positive | Positive |
| Positive | Negative | Zero |
| Positive | Zero | Positive |
| Positive | Positive | Large Positive |

### 5.4. Neural Network Controller

The neural controller developed and exploited in this research exhibits properties similar to proportional-derivative action. The developed controller is based on the structures presented in References [31–34], realizing a control action similar to PD. The use of three inputs, where one of them is the derivative of the control error, removes the problem of necessity of implementation of memory block within the network structure. However, the neural network controller, contrary to PID or fuzzy, is not a standard solution for the final control elements.

As a rule, the available computational power of real positioners is strongly limited. Therefore, the simple, unidirectional straightforward neural network structure was chosen in order to speed up the processing time. Moreover, the use of a unidirectional network facilitates the optimization

of weights. The two-layer neural network structure with a hidden layer counting 10 neurons was designed. A sigmoid activation function is applied for each neuron. The network was trained by means of the back-propagation algorithm. The structure of applied neural network is presented in Figure 11.

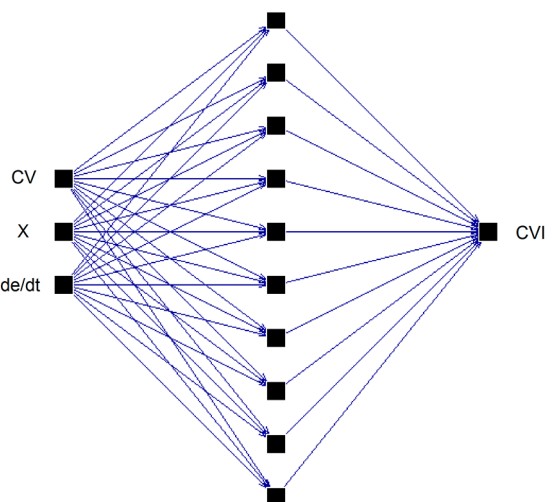

**Figure 11.** The structure of the neural network controller NN.

The block diagram of the Simulink model of neural network controller is depicted in Figure 12.

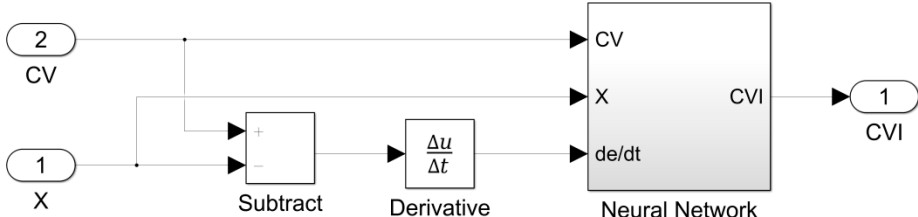

**Figure 12.** The simplified block schematics of the Simulink model of the neural PD controller.

## 6. The Choice of Simulation Tests

Let us briefly discuss operational conditions of the investigated final control element in order to choose the set of its components liable to accelerated wear. This would be assumed as justifying to some extent the rational choice of the primary set of performed simulation tests. Let us firstly define the list of harsh working conditions of the final control element:

- chemically aggressive and corrosive environment,
- leakages of the controlled aggressive media,
- high operating temperature,
- environmental pollution,
- control loop oscillations,
- mechanical impacts,
- vibrations.

The two exemplary snap shots approaching real operational conditions of the final control elements are presented in Figure 13.

The corrosion, mechanical impacts as well as dusty environment influences friction force in control valve packing. The friction may be assumed as the root cause of the hysteresis loop visible in Figure 7. This hysteresis may significantly expand or shrink during the exploitation period. The positioner of the

final control element should compensate for the effects of changes of the hysteresis of the pneumatic actuator. Therefore, it is justified to investigate the influence of the bi-directional changes of the friction of the controller effort.

(**a**)                    (**b**)

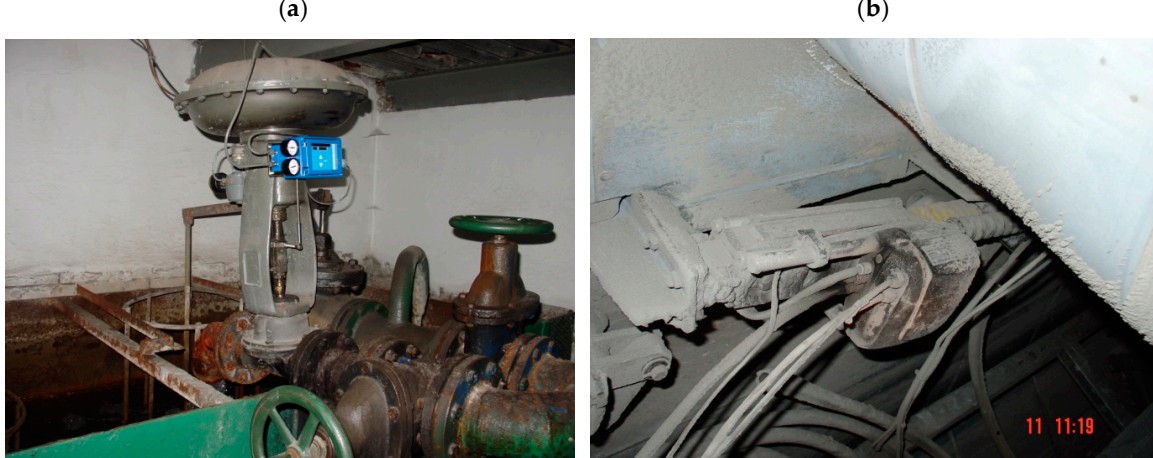

**Figure 13.** (**a**) Snap shot of the final control element installed in the liquid level control loop. The effects of a highly corrosive environment are visible by naked eye. The ambient temperature varies in the range 15–50 °C, while relative humidity in the range 50–95%; (**b**) An example of the application of the positioner for the control of air to fuel ratio in one of the black coal power stations. The final control element has been installed in extremely hot and dusty environments.

Moreover, the chemically aggressive environment is not inactive in regard to the actuator's spring. The spring is surrounded by an ambient atmosphere in the single acting actuators. Even tiny shrinkage of the diameter of the spring's wire caused by a corrosion seriously influences the spring's elasticity. The reason is that spring elasticity depends on the fourth power of the wire diameter. This justifies the experiments were the actuator's spring elasticity drops down. On the other hand, there is imaginable a replacement of the worn spring by a new one. This also justifies the experiments where the spring elasticity could be greater than nominal.

The tests of internal positioner controller were carried out for:

- nominal values of friction and actuator's spring elasticity;
- friction varying within the range [−50%, +50%];
- actuator's spring elasticity varying within the range [−50%, +50%].

In order to minimize the simulation time, the tests were performed only for the extreme values of the above-mentioned influencing quantities.

## 7. Discussion of Obtained Results

The collected quality factors values obtained in the frames of performed simulation tests are presented in Table 3. The best results are highlighted in bold.

### 7.1. Control Effort

Table 3 provides interesting data for discussion. In comparison to the obtained results for different types of examined controllers, it is possible to admit that classic proportional-and-derivative controllers might be the best choice for the positioner. It provides almost the best values of control quality factors and acceptable values of controller effort. It seems that the fuzzy controller serves even slightly better figures. However, the implementation of the fuzzy controller is much more burdensome than classic PD. In turn, the neural network controller provides worse control quality factors beyond the controller effort. Definitely the worst quality control factors were obtained in all categories for the proportional

controller. The examples of response of the controllers having the highest effort are shown in Figure 14. The outputs were the yield for the 0.0125Hz trapezoidal set-point with the slope equal to 0.025 s$^{-1}$. In contrast, Figure 15 depicts the best results achieved for classic proportional-and-derivative and fuzzy logic controllers.

**Table 3.** Experimentally obtained values of control quality factors.

| Experiment | | $Q_K$ | $e_{Kr}$ | $e_{Ks}$ | $\kappa_{rise}$[%] | $\kappa_{fall}$[%] | $T_{Rrise}$[s] | $T_{Rfall}$[s] |
|---|---|---|---|---|---|---|---|---|
| | | | | | **Quality Factor** | | | |
| **P controller** | Nominal parameters | 0.397 | 4.03 | 9.38 | 8.67 | 0.79 | 8.67 | 29.2 |
| | Reduced friction | 0.402 | 3.98 | 9.36 | 2.62 | 0.77 | 8.65 | 29.2 |
| | Increased friction | 0.381 | 4.04 | 9.50 | 2.69 | 0.79 | 8.75 | 29.4 |
| | Reduced elasticity | 0.428 | 3.92 | 8.84 | 3.41 | 0.99 | 7.43 | 27.3 |
| | Increased elasticity | 0.344 | 4.34 | 10.2 | 2.12 | 0.64 | 10.1 | 31.0 |
| **PD controller** | Nominal parameters | 0.127 | 3.26 | **8.81** | 0.06 | **0.00** | **7.14** | **29.2** |
| | Reduced friction | 0.190 | 3.23 | **8.79** | 0.06 | **0.00** | **7.14** | **29.2** |
| | Increased friction | 0.055 | 4.79 | **9.37** | 0.04 | **0.00** | **7.20** | **29.6** |
| | Reduced elasticity | 0.288 | 2.94 | **7.86** | 0.05 | **0.00** | **5.79** | **27.3** |
| | Increased elasticity | 0.104 | 3.72 | **9.78** | 0.05 | **0.00** | **8.62** | **30.9** |
| **Fuzzy PD controller** | Nominal parameters | 0.107 | **3.25** | 8.83 | **0.05** | 0.20 | **7.14** | **29.2** |
| | Reduced friction | 0.155 | **3.22** | 8.81 | **0.04** | 0.19 | **7.14** | **29.2** |
| | Increased friction | 0.046 | **4.51** | 9.34 | **0.02** | 0.00 | **7.20** | **29.3** |
| | Reduced elasticity | 0.150 | **2.91** | 7.90 | **0.04** | 0.29 | **5.78** | **27.3** |
| | Increased elasticity | 0.086 | **3.71** | 9.76 | **0.04** | 0.13 | **8.62** | **30.9** |
| **Neural NN controller** | Nominal parameters | **0.048** | 7.06 | 14.77 | 0.39 | **0.00** | 7.37 | >50 |
| | Reduced friction | **0.043** | 7.05 | 14.73 | 0.36 | **0.00** | 7.37 | >50 |
| | Increased friction | **0.049** | 7.11 | 14.84 | 0.43 | **0.00** | 7.38 | >50 |
| | Reduced elasticity | **0.122** | 6.64 | 14.45 | 1.27 | **0.00** | 5.99 | >50 |
| | Increased elasticity | **0.041** | 7.64 | 15.06 | 0.30 | **0.00** | 8.91 | >50 |

It is clear to see from Figure 14 that lowest control quality controllers generate a significant number of high amplitude oscillations. On the other hand, the outputs of the FPD and NN controllers presented in Figure 15 are characterized by a relatively low number of strongly damped oscillations. This should be considered as highly beneficial from the perspective of elongation of the life-time of the electro-pneumatic transducer.

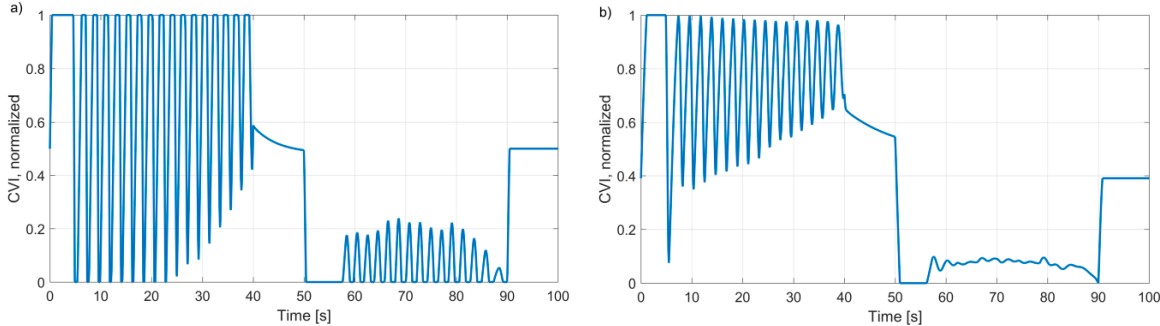

**Figure 14.** (**a**) Output of P controller and (**b**) output of PD controller.

The most important results of the performed research are shown in Figures 16–19. The mean time efforts for all controllers are shown in Figure 16. The mean time effort was calculated here in 10 s long moving time windows. From this figure, it comes that the mean-time effort in short-term vary significantly for studied controllers. Of course, the short-term mean-time effort does not reflect the long-term development of control effort. Much more convenient, in the longer time perspective, is to track the cumulative effort over a longer time. This is depicted in Figure 17.

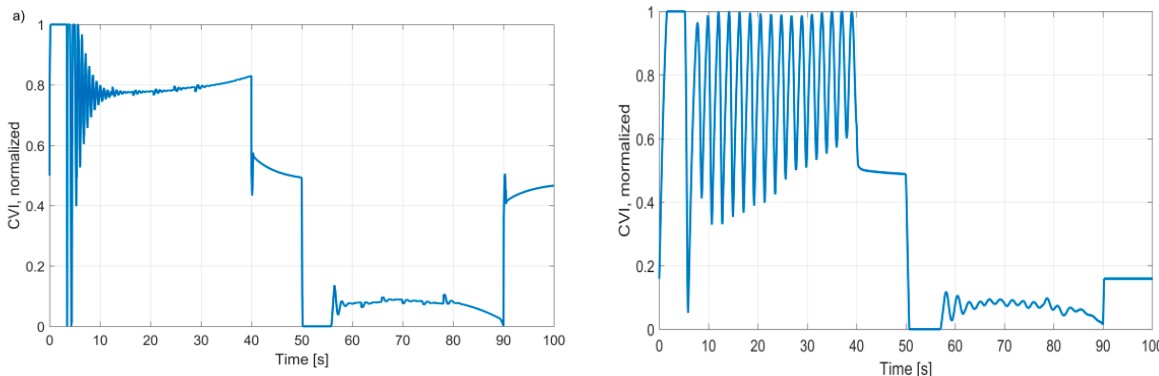

**Figure 15.** (**a**) Output of fuzzy logic FPD controller and (**b**) output of neural network NN controller.

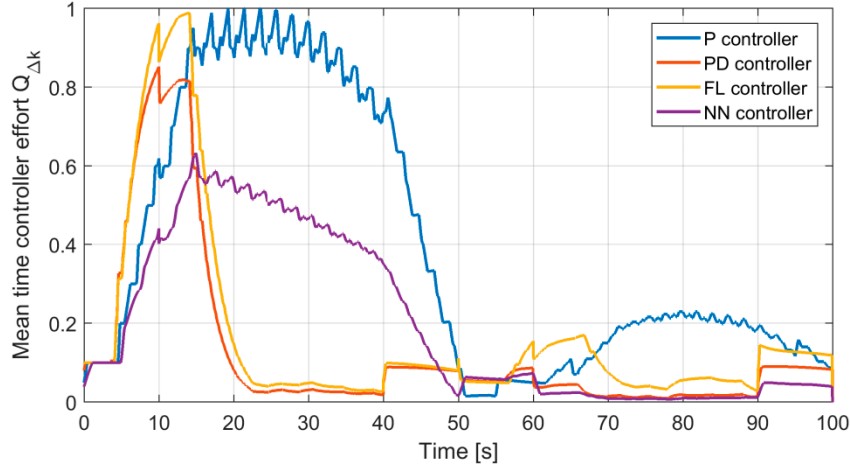

**Figure 16.** The comparison of the mean time effort of the four investigated controllers ($\Delta T = 10$ s).

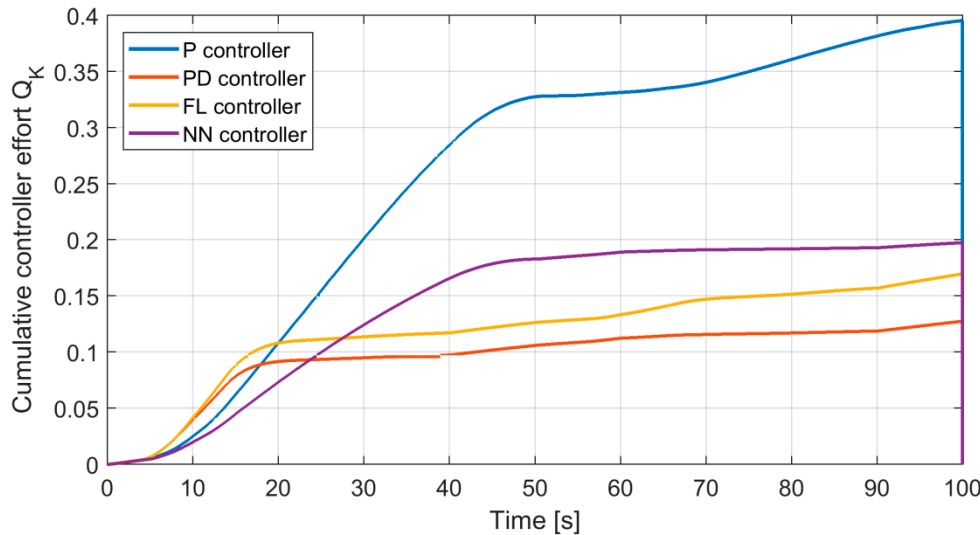

**Figure 17.** The cumulative effort of the four investigated controllers in time horizon $T = 100$ s.

The cumulated effort was calculated for the time horizon $T$ varying from 0 to 100 s. This factor allows for the easiest and fastest evaluation of the choice of the proper structure of the internal controller in terms of minimizing the effort and energy consumed from the auxiliary air supply source (Figure 3). Figure 17 shows that in time window (up to 25 s), the cumulative effort of

proportional-differentiating controllers is higher than for proportional controllers. However, from a long-term perspective, the situation evidently changes.

It is also important to determine how the choice of the control strategy affects the durability and fatigue of the electro-pneumatic transducer. The durability can be characterized indirectly by the distribution of amplitude-cycles (A–S) curve. The appropriate graph for the four tested controllers is depicted in Figure 18. This graph was achieved by usage of chirp shaped set-points. This graph shows the results that derivative action generates a large number of cycles with relatively small amplitude, and a relatively low number of high amplitude cycles. This observation allows further consideration about the application of an additional filter for damping small amplitude cycles. Quasi-constant A-S curve for proportional action controller gives an assumption to forecast earlier fatigue of the driven electro-pneumatic transducer. It is also clear to see from Figures 17 and 19 that, in the case of chirp set-point, the cumulative effort of derivative action controllers is also significantly smaller.

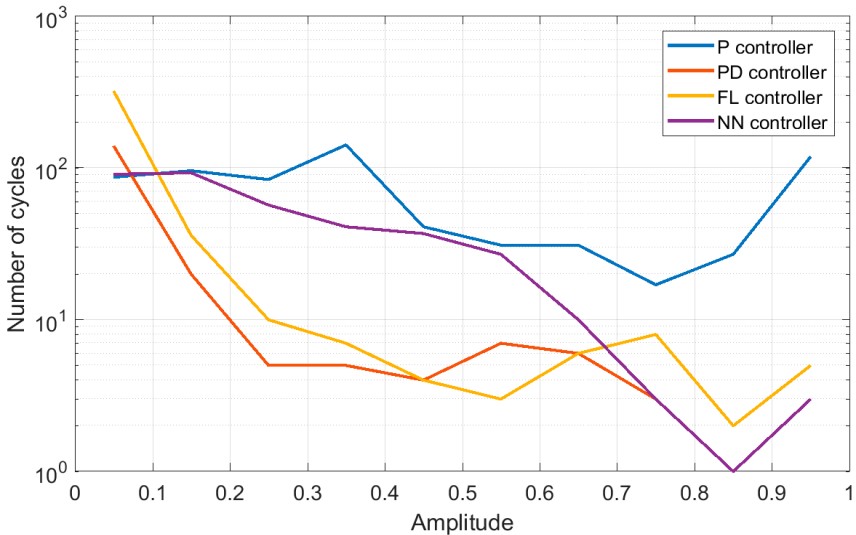

**Figure 18.** Amplitude-cycles curve. Conditions: setpoint—chirp signal; observation time $-1000$ s.

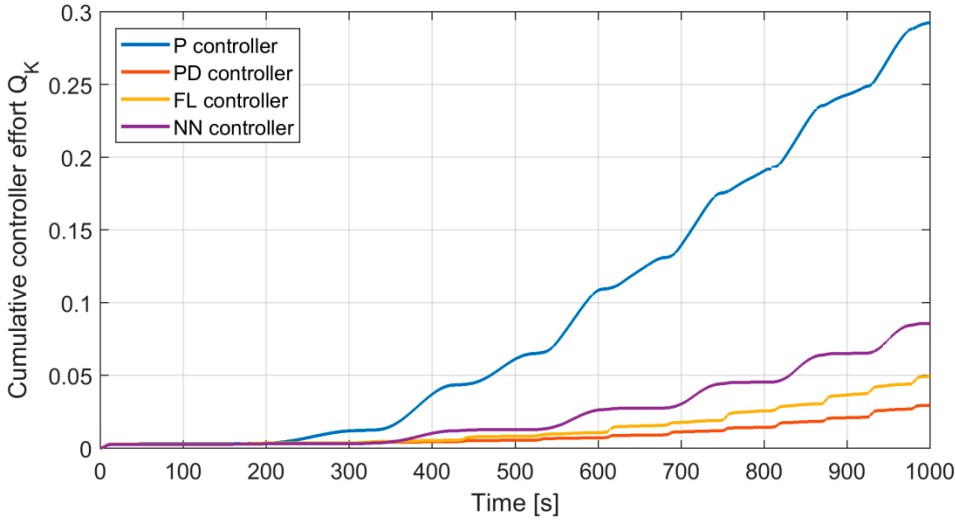

**Figure 19.** The cumulative effort of the four investigated controllers in 1000s time horizon by chirp set-point.

### 7.2. The Low Pass Filter Versus Control Effort

It seems that the simplest way of diminishing the control effort is to apply a low pass filter following the output of positioner controller. Obviously, this influences the dynamics of the whole control loop. The series of simulations were made in order to find the consequences of applying the controller output filter. The simple first order lag filer (12) was used.

$$G(s) = \frac{1}{T_f s + 1} \tag{12}$$

The obtained results of the cumulative effort versus time constant $T_f$ are shown in Figure 20.

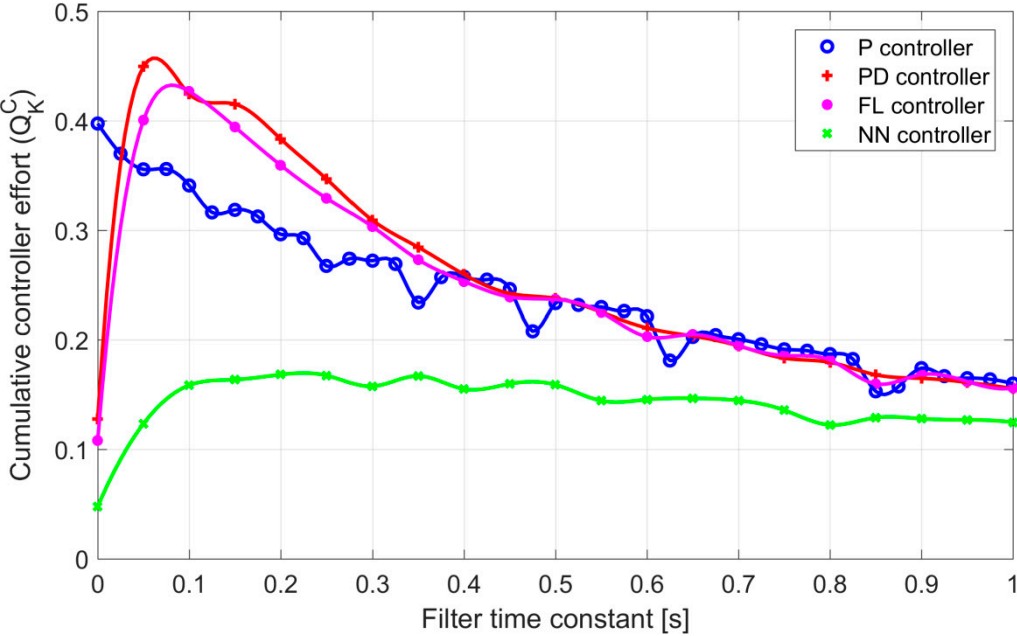

**Figure 20.** Cumulative effort of the investigated controllers versus filter time constant.

Unexpectedly, the cumulative effort does not immediately fall down, may be with exception of the P controller. Indeed, the application of the filter may even significantly increase the effort for PD, FPD, and NN controllers. This effect can be explained by the low pass frequency characteristics of applied filter. It is recognized particularly for the low values of the filter time constant. Interestingly, the cumulative effort for all analyzed controllers converges to some relatively low level. This justifies the supposition that applying the low pass filter will lower the cumulative effort. Obviously, it is important to track how the application of such filter will influence other control quality factors. Figure 21 shows simulations of cumulative effort and tracking errors $e_{Kr}$ and $e_{Ks}$ versus the filter time constant.

All charts in Figure 21 show that application of the low pass filter does not bring any benefits in respect to tracking errors $e_{Kr}$ and $e_{Ks}$. Inversely, the usage of low pass filter evidently worsen tracking properties of the control system. It is very interesting also note that approximately proportional increase of tracking errors versus filter time constant. As the trends of cumulative effort and tracking errors versus the filter time constant are contrary, it is clear that a space for a trade-off between controller effort and tracking error exists.

Let us now discuss how the filtering of the controller output signal influences the remaining control quality factors. Figure 22 depicts the impact of filtering on settling times and overshoots.

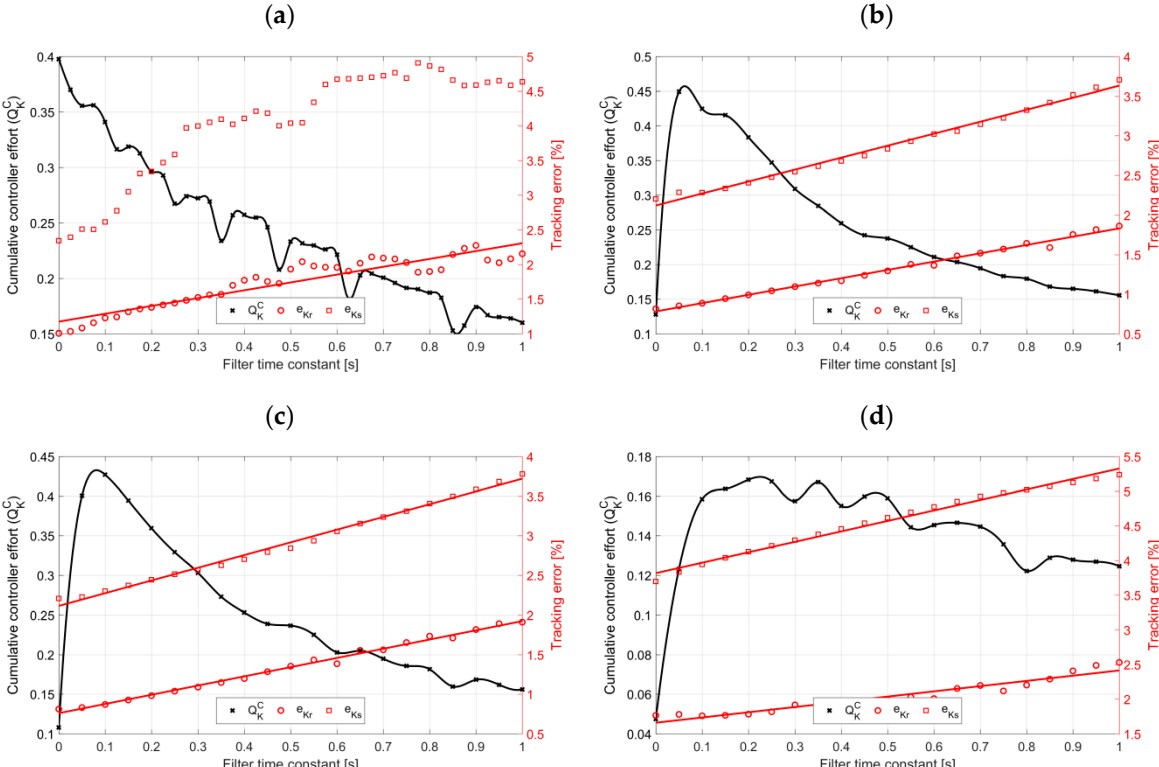

**Figure 21.** Cumulative effort and tracking errors $e_{Kr}$ and $e_{Ks}$ versus filter time constant for: (**a**) P controller, (**b**) PD controller, (**c**) FPD controller, and (**d**) NN controller.

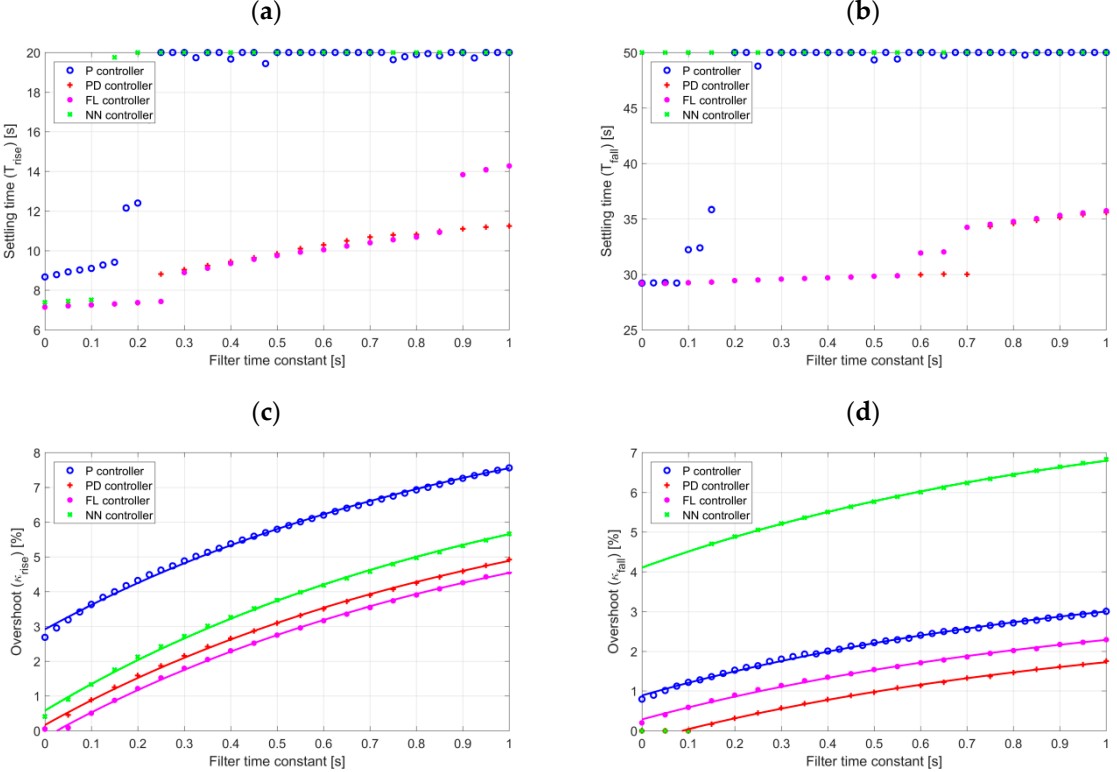

**Figure 22.** Control quality factors versus filter time constant. (**a**) settling time $T_{rise}$, (**b**) settling time $T_{fall}$, (**c**) overshoot $\kappa_{rise}$, and (**d**) overshoot $\kappa_{fall}$.

From Figure 22, clearly follows that application of a low pass filter does not seem to be justified in terms of keeping low values of considered control quality factors. From this figure, one can observe interesting exponential regularity of the overshoot.

$$\kappa_{rise/fall} = a \cdot e^{-T} + b \qquad (13)$$

where gain coefficients *a* and offsets *b* are given in Table 4.

**Table 4.** Coefficients of approximations of overshoots versus filter time constant T.

| Controller | Overshoot | *a* | *b* |
|:---:|:---:|:---:|:---:|
| P | $\kappa_{rise}$ | −7.33 | 10.3 |
|  | $\kappa_{fall}$ | −3.35 | 4.24 |
| PD | $\kappa_{rise}$ | −7.47 | 7.63 |
|  | $\kappa_{fall}$ | −3.14 | 2.88 |
| FPD | $\kappa_{rise}$ | −7.37 | 7.23 |
|  | $\kappa_{fall}$ | −3.17 | 3.45 |
| NN | $\kappa_{rise}$ | −8.03 | 8.61 |
|  | $\kappa_{fall}$ | −4.25 | 8.36 |

If the aperiodic step response is required then the overshoots are not allowable. If we look at the charts (c) and (d) of Figure 22, the space for trade-off between controller effort and filter time constant seems to be extremely narrow.

## 8. Summary

The three practical measures of the control quality have been proposed in this paper, namely: Variability, mean time, and the cumulative effort. These measures can be applied for the design of a cost function, which allows for optimization of positioner controller settings with respect to the controller effort. Minimization of the controller effort would be beneficial because the lower effort the lower wear of moving parts of positioner is, the lower the energy (compressed air) consumption.

On the example of a fairly typical liquid level control system, it was shown (Table 3) that the replacement of positioner controller from classic P algorithm to PD or fuzzy PD allows more than a three-fold reduction of effort, while simultaneously obtaining much better values of other quality control factors. It is worth mentioning that frequently the P controller is preferred in positioners. This comes from considerations regarding the cascade system where the external controller performs integral action and internal controller proportional action. As follows from the results of investigations presented in this paper, it does not promise either longer lifetime nor better control quality.

As shown in the paper, the diminishing of the control effort by means of low pass filter of the control output is possible, however, may significantly worsen tracking properties of the control system. This applies to settling times and overshoots as well. Therefore, the use of low pass filter of the control signal should be considered as problematic.

The simulation tests demonstrate that by the proper selection of the structure and parameters of the internal controller of the final control element, one can achieve two seemingly contradictory outcomes. On the one hand, better quality control, while on the other, simultaneous reduction of the controller effort. Obviously, the experimental verification of simulation findings is necessary. This is foreseen for the near future.

**Author Contributions:** Conceptualization, M.B.; Data curation, B.H.; Formal analysis, M.B.; Investigation, B.H.; Methodology, M.B.; Project administration, M.B.; Resources, B.H.; Software, B.H.; Supervision, M.B.; Visualization, M.B. and B.H.; Writing—original draft, M.B.; Writing—review & editing, M.B.

**Funding:** This research received no external funding.

**Conflicts of Interest:** The authors declare no conflict of interest. The funders had no role in the design of the study; in the collection, analyses, or interpretation of data; in the writing of the manuscript, or in the decision to publish the results.

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
