# Peer review of "The Trade-Off between the Controller Effort and Control Quality on Example of an Electro-Pneumatic Final Control Element"

_actuators, doi:10.3390/act8010023_

Round 1
Reviewer 1 Report
Originality/novelty:
The authors are proposing the shaping and tuning of the investigated final control element for reducing the wear, fatigue and failure of its moving parts, while keeping the controlled system's performance at the same levels. Although the analysis of the controller effort and the investigated controllers are not novel contributions, the overall originality of the concept and utilized methodology is sufficient.
Technical and Scientific Soundness:
The proposed methodology is sound, although evaluated only through simulations. Experimental evaluation and the use of additional filtering would provide more concrete information on how the quality control factors are affected. Although the dynamic affect of adverse environmental conditions is commented in detail in the Introduction, their incorporation into the simulation and their effect is not clearly analyzed. The included nonlinearity and asymmetry in the static and dynamic characteristics are sufficient for this evaluation but this reviewer would suggest more comments on the effect of adverse conditions.
Quality of presentation:
Clarity of presentation and overview of the analyzed concepts is high. The information and methods are adequately presented, while the discussion of results providing clear points to support the authors claims.
Author Response
Dear Reviewer,
please find answers to the review in attached file.
With best regards
Michal Bartys

Reviewer 2 Report
1. The main objective of the article is to demonstrate the best control among four. But it is not justified because only these four were selected.
1. The main objective of the article is to demonstrate the best control among four. But it is not justified because only these four were selected. But there are several control models: genetic algorithm immune, PID, adaptive, etc
2. In the summary it says "In this paper, a practical and economic approach to prolongation of final control element lifetime is proposed." But it does not have its verification in the article neither practical nor economic only with theoretical modeling.
3. In the introduction, the analyzes of the articles on these four types of control were not presented.
4. In the models and methods, you must present enough information for its repetition. But in the modeling of the system it was not presented:
a. data of the elements of the system:
i. parameters, characteristics and where they are installed, what their relationship.
ii. Elasticity and friction data of the elements.
b. Data of the control systems:
i. If it is P controller; o PD controller coefficients
ii. If it is fuzzy logic based controllers or of neural network controller its characteristics and parameters.
c. Schemes, types and / or characteristics of the sensors
5. The authors cite their work [9], but in this article only the PID control is investigated and they do not present the characteristics of other types of controls.
6. It is not clear for which parameters (static and dynamic) of the elements of the system, characteristics of the control, the results appear in the table (these results can not be repeated in another investigation). Only presented to pneumatic actuator (Formula 7) without explaining in which part this formula is used.
7. The time interval - T is not justified. The T value at which performance plateaus depends on the plant dynamic characteristics. ( Watch this - https://www.mathworks.com/help/mpc/ug/choosing-sample-time-and-horizons.html )
8. Some statements are not justified in references such as:
a. “Therefore, the final control elements are classified as belonging to the group of elements of automation systems subjected to the most frequent failures”
b. “It is estimated that among instrumentation, actuators and technological components, the share of failures of final control elements exceeds 45%.”
c. “which is recognized widely in the field of diagnostics and process safety”
9. Some of his appointments are not well referenced [6], (MAIN APPOINTMENT). It is an Internet site that has many pages and manuals. It is impossible to find the required information.
10. There are errors in the text. The authors must verify the writing well.
11. See the idea (example) how to present this type of research in - Chun-Tang Chao, Nana Sutarna, Juing-Shian Chiou and Chi-Jo Wang Equivalence between Fuzzy PID Controllers and Conventional PID Controllers, Appl. Sci. 2017, 7, 513; doi:10.3390/app7060513
12. In the text there is no justification for these conclusions:
This is of great practical importance because the failures of electro-pneumatic transducers are very prevalent in electro-pneumatic positioners. Additionally, it also has economic benefits because it reduces the energy consumption by the actuators.
It is worth to mention, that frequently P controller is preferred in positioners. This comes from considerations regarding cascade systems where the external controller performs integral action and internal controller proportional action. As follows from the results of investigations presented in this paper, it does not promise neither longer lifetime nor better control quality.
13. This is not the conclusion:
It would be advisable in the future to investigate the use of additional controller output filtering to further reduce the actuator effort.
Author Response

(The authors gave the same response as above.)

Reviewer 3 Report
The authors discuss the results of a control structure with different control parameters, trying to reduce the control effort and to increase the performance outputs. The topic is for general interest for control engineers, but I am not sure if the content is for special interest for this journal. The reference list must be updated with control structure references. The language is proper. The paper structure is not ideal. From a whole of 12 pages length, 6 pages are used to discuss general notions, definitions, well-known issues. Moreover, the presented general notions are not used. For example it is discussed a classical control block diagram (Figure 1) and it is used a cascade control structure (Figure 5). Or it is presented the block diagram of an electro-pneumatic final control element, but it is not used in the research. The added value of the paper is not clear. Why is it important to present equation (7)? Or if it presented, please detail how did you identified these transfer functions or give the corresponding reference. Are presented the results for different controllers. Which are the parameters of these controllers? How were these controllers designed? And first of all, which is the model of the process you used for design? The process is not even presented, we known only that it is a tank (no characteristics). What kind of performance measures were used for controller design? Which is the cost function in the neural network? Which are the parameters of your neural network (number of layers, number of cells, etc.)? All these parameters affect both the control effort and performance of the system, so your conclusions are not relevant if these aspects are not discussed. For example including in the cost function of the neural network the minimization of control effort, you can obtain great results without increasing or decreasing the friction of the actuator. Which is the conclusion for the reader? How to adjust the actuator in the future to obtain the best control? It is important to offer a general conclusion, general rule for this, not a conclusion for this particular case study.
Author Response

(The authors gave the same response as above.)

Round 2
Reviewer 2 Report
The article has improved a lot and can be published as is.
Author Response
Dear Reviewer,
I would like to thank you for your valuable remarks and co-operation in the process of improvement of the quality of the paper.
With best regards
Michal Bartys
Reviewer 3 Report
The revised paper is more clear than the original, but it is still not in a publishable form. The basic issues are still not solved. A comparison of a PID controller with a fuzzy and neural network controller is not fair. Are imposed different performances to design these controllers, so we can not expect to have the same results. Compare the control effort for two controllers designed in the same way. In present form the conclusions are not relevant.
Author Response
Dear reviewer,
please find answers to remarks in attached file.
With best regards
Michal Bartys

Round 3
Reviewer 3 Report
Although I still consider that a comparison between different type of controllers (designed for different performance measures) is not fair, the paper is now in a publishable form.